# Wikidata e Thesaurus Nuovo soggettario: insieme per costruire un'ontologia di dominio nell'ambito della fotografia

Silvia Bruni, Elena Cencetti, Alida Daniele, Valentina Lepore, Fabrizio Nunnari, Elisabetta Viti

Il presente contributo offre una panoramica sulla collaborazione tra il Gruppo Wikidata dell'Università di Firenze, alcuni bibliotecari del Settore Ricerche e strumenti di indicizzazione semantica della Biblioteca nazionale centrale di Firenze (BNCF) e Wikimedia Italia, con l'obiettivo di sperimentare la creazione in Wikidata di un'ontologia di dominio sulla fotografia,  con un colloquio semantico e tecnico con il Thesaurus Nuovo soggettario.

Il dialogo tra i due strumenti è possibile grazie ai dati strutturati e aperti del Thesaurus, conformi agli standard internazionali e ai FAIR principle di ricercabilità, accessibilità, interoperabilità e riusabilità.

Costruire un'infrastruttura di lavoro con Wikidata è stata una sfida che ha tenuto conto di:

1.  dimensione strutturale e funzionale dei due strumenti;

2. area culturale,  utenza e *mission* delle istituzioni che hanno partecipano al progetto;

3. poliedricità, specificità, monoreferenzialità, modelli comunicativi della terminologia specialistico-settoriale analizzata;

Dall'analisi dei due strumenti, diversi per natura, struttura e contesto sono emerse criticità dovute a un disallineamento semantico che ha portato ad una revisione in entrambi i database di significanti e significati.

## Altre publicazioni pertineti al tema degli autori:

☐  Cencetti, E., Viti, E., & Pelizzari di San Girolamo, C. (in press). Termini, dati e collegamenti: 'Conversazioni' tra il Thesaurus del Nuovo soggettario e Wikidata. *Imagines delle Gallerie degli Uffizi*.

☐  Bruni, S., Materassi, L., Martone, C., & Secinaro, E. (2020). Wikipediani, utenti remoti e "operosi" di servizi bibliotecari. *Biblioteche oggi trends, 6*(1).

☐  Bruni, S., Calonaci, B., Giaccai, S., Margheri, M., & Viti, E. (2018). Wikipedia (and his brothers) to network and improve services. *JLIS, 9*(3).
