# OpenReview forum: "Wikidata e Thesaurus Nuovo soggettario: insieme per costruire un’ontologia di dominio nell’ambito della fotografia"
_wikimedia.it/Wikidata_and_Research/2025/Conference — WD&R Poster_

### Official Review · ~Alessandra_Boccone1 · 2025-01-07
**Ottima proposta**

**Originality:** 5
**Impact:** 5
**Confidence:** 5

**Review:**

La peculiarità del progetto presentato e la molteplicità dei soggetti coinvolti denotano l'alta qualità della proposta, che colpisce anche per i possibili sviluppi futuri. L'allineamento fra Wikidata e il Nuovo Soggettario nella creazione di un'ontologia per la fotografia, con la necessaria revisione di entrambi i database, denota un approccio critico e costruttivo, base essenziale per l'evoluzione di entrambi gli strumenti.

**Compliance:**

5

**Final Paper Review:**

Interesting poster, great idea the inclusion of the example that helps to better understand the working methodology. Great work!

**Scientific Quality:**

5

---

### Official Review · ~Alessio_Melandri1 · 2025-01-21
**Bridging ontologies**

**Originality:** 3
**Impact:** 4
**Confidence:** 4

**Review:**

This proposal highlights a noteworthy interdisciplinary collaboration to create a domain ontology for photography in Wikidata. The integration emphasizes adherence to FAIR principles, enhancing the interoperability and reusability. The discussion of semantic misalignments and their resolution demonstrates a nuanced understanding of the challenges posed by specialized terminology. While the focus is impactful, the paper could expand on the methods used to overcome technical and cultural differences between the systems. Nevertheless, this project provides a strong foundation for further ontology development and improved knowledge representation in Wikidata.

**Compliance:**

5

**Final Paper Review:**

Strengths
1) Clear objective and cultural relevance: The poster targets a meaningful and culturally valuable goal: enhancing semantic interoperability between a national authority vocabulary and an open, collaborative global platform. Focusing on the domain of photography provides a concrete and bounded context for experimentation.
2) Practical contribution through ontology alignment: The authors describe a clear example of how discrepancies between ThNS and Wikidata (e.g., lack of a specific entity for individual photographs) are resolved through the creation of new entities and mappings. This reflects a sound, practical understanding of ontology building.

Weakness
1) Lack of evaluation or scalability insight:
While the poster successfully communicates the conceptual alignment process and the manual curation of terms, it lacks any mention of how this methodology could scale. Include quantitative or process data (e.g., number of potential new entities created, alignment effort in hours or terms).

**Scientific Quality:**

4

---

### Decision · Program_Chairs · 2025-02-05

Accept (Poster)